

# Exact finite-size scaling of the maximum likelihood spectra in the quenched and annealed Sherrington-Kirkpatrick spin glass

Ding Wang[1,2] and Lei-Han Tang[1,3⋆]

**1** Department of Physics, Hong Kong Baptist University,
Kowloon Tong, Hong Kong SAR, China
**2** Department of Materials Science and Engineering,
Southern University of Science and Technology, Shenzhen, China
**3** Center for Interdisciplinary Studies, Westlake University,
Hangzhou 310024, Zhejiang, China

⋆ tangleihan@westlake.edu.cn

## Abstract

Fine resolution of the discrete eigenvalues at the spectral edge of an $N \times N$ random matrix is required in many applications. Starting from a finite-size scaling ansatz for the Stieltjes transform of the maximum likelihood spectrum, we demonstrate that the scaling function satisfies a first-order ODE of the Riccati type. Further transformation yields a linear second-order ODE for the characteristic function, whose nodes determine leading eigenvalues. Using this technique, we examine in detail the spectral crossover of the annealed Sherrington-Kirkpatrick (SK) spin glass model, where a gap develops below a critical temperature. Our analysis provides analytic predictions for the finite-size scaling of the spin condensation phenomenon in the annealed SK model, validated by Monte Carlo simulations. Deviation of scaling amplitudes from their predicted values is observed in the critical region due to eigenvalue fluctuations. More generally, rescaling the spectral axis, adjusted to the distance of neighboring eigenvalues, offers a powerful approach to handling singularities in the infinite size limit.

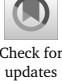

# 1 Introduction

A recent paper by Foini and Kurchan [1] rekindled interest in the annealed spin-glass and related models where interactions between the spins, for example, are allowed to evolve according to the rules of equilibrium thermodynamics. At sufficiently low temperatures, spins and interactions conspire to yield self-planted states whose spin fluctuations in each instance resemble those of the Mattis model, a ferromagnet in disguise [2–4]. However, the translational or permutation symmetry of the annealed model implies an exponential number of such states, with no discernible order from spin configurations alone. As the interactions drift over time, each self-planted spin state with "hidden Mattis order" evolves accordingly, with memory of its previous configurations determined by the rate of change of the coupling matrix. As such, these annealed models may be considered as prototypes of certain glass forming systems that lack identifiable structural regularity yet behave dynamically as a solid or an extremely viscous fluid. In biological systems, interactions among proteins and other cellular components can be reset via chemical modifications such as phosphorylation and acetylation, etc [5,6]. These could then change the way proteins bind each other to form condensates or other spatial structures through quasi-equilibrium processes.

The mean-field Sherrington-Kirkpatrick (SK) model offers a concrete example where the above-mentioned phenomenon can be examined quantitatively in the random matrix formulation. Here, onset of the Mattis-type order is identified with gap opening at the leading edge of the coupling matrix spectrum. The associated principal eigenvector then specifies the instantaneous hidden order in the spin configurations. By integrating out the spin degrees of freedom, spectral properties of the coupling matrix can be analyzed in a one-dimensional Coulomb gas formulation. Interestingly, the annealed or probabilistic Coulomb gas formulation can also be used to learn about various statistical properties of a class of matrix models, where the distribution of the elements are correlated with each other in some way [1]. By linking the annealed problem to the theory of random matrices, many powerful tools developed in the latter community can be employed to generate precise mathematical results.

The aim of this paper is to extend the pedagogical discussions by Foini and Kurchan to systems of finite size, particularly near the critical point for gap opening. To achieve this, it is essential to resolve the spacing between successive eigenvalues of the coupling matrix at the spectral edge, which, surprisingly, has not been systematically addressed despite many closely related work and results (see, e.g., [7–10]). Focusing on the maximum likelihood spectrum, we develop a novel procedure based on its Stieltjes transform. By applying a scaling ansatz, we obtain a first order Riccati-type ODE, which can be transformed into a second-order linear ODE. For the original random matrix, the solution is given by the Airy function, with its nodes indicating positions of successive eigenvalues near the spectral edge. More generally, the edge spectrum of the matrix can be determined through a numerically exact procedure based on our formulation. These results provide a complete description of the onset of the hidden Mattis order in the annealed SK model in a large but finite system.



Beyond the annealed SK model, our approach may offer insights into finite-size effects in a broader range of matrix models. Random Matrix Theory (RMT) has established itself as a fundamental framework for understanding the statistical properties of complex systems. Originally developed to model the energy levels of heavy nuclei [11], RMT has since been widely applied across various fields, including quantum physics [12, 13], biology [14, 15], and finance [16, 17]. A common aspect among these applications is the use of finite-sized matrices, where size effects are critical and can substantially influence theoretical predictions and practical interpretations.

The paper is organized as follows. In Section 2, we introduce three variants of the SK model and present a heuristic discussion of their thermodynamic relations using a generic phase diagram for spin glasses. We briefly review the mapping to the spectral representation of the quenched and annealed models, along with their respective Coulomb gas energy functions. Section 3 explores the maximum likelihood solution of the coupling matrix spectrum for finite-sized systems, with particular emphasis on the largest eigenvalues at the spectral edge. Their precise location is resolved through a perturbative procedure in $1/N$. In Section 4, we present finite-size scaling predictions for the principal and sub-dominant spin fluctuation amplitudes for both quenched annealed models across the transition temperature. These scaling relations are verified through Monte Carlo simulations of the annealed SK model. Noticeable discrepancy in scaling amplitudes against the maximum likelihood estimations is observed in the critical region.

## 2 The SK model and its spectral representation

### 2.1 Quenched, annealed, Mattis models and the Nishimori line

The Sherrington-Kirkpatrick spin-glass model [18] is originally introduced for $N$ Ising spins $S_i = \pm 1$ with pair-wise interactions given by the Hamiltonian,

$$H(\{S_i\}, \{J_{ij}\}) = -\sum_{i<j} J_{ij} S_i S_j, \tag{1}$$

where the coupling constants $J_{ij} = J_{ji} = \pm 1/\sqrt{N}$ with equal probability. The free energy of the quenched model at temperature $T = 1/\beta$ is given by $F_q = -T \ln Z_q$ where

$$Z_q(\{J_{ij}\}) = \sum_{\{S_i\}} \exp[-\beta H(\{S_i\}, \{J_{ij}\})], \tag{2}$$

is the partition function for a given configuration of the $J_{ij}$'s. As is well known, the model has a phase transition at $T_q = 1$, below which the spin-glass order develops.

Compared to the quenched model, the "annealed partition function"

$$Z_a = \sum_{\{S_i\}} \left\langle \exp[-\beta H(\{S_i\}, \{J_{ij}\})] \right\rangle_{\{J_{ij}\}} = 2^N [\cosh(\beta N^{-1/2})]^{N(N-1)/2}, \tag{3}$$

is much easier to calculate. The corresponding free energy free energy $F_a = -T \ln Z_a$, on the other hand, is a non-monotonic function of $T$ and gives rise to a negative entropy $S_a = -\partial F_a / \partial T$ at sufficiently low temperatures. On the other hand, if we consider the $J_{ij}$'s as another set of $N_b = N(N-1)/2$ state variables, the "entropy crisis" disappears, with the proper partition function given by $\tilde{Z}_a = 2^{N(N-1)/2} Z_a$. In fact, the ground state is now $2^N$ fold degenerate with a zero temperature entropy $\tilde{S}_a(T=0) = N \ln 2$.

In 1976, Mattis [2] considered the case $J_{ij} = \frac{1}{N}\sigma_i\sigma_j$, i.e., the coupling constants are not chosen independently, but instead constructed from a "hidden" spin configuration $\{\sigma_i\}$. A simple gauge transformation $S_i \to S_i\sigma_i$ then brings Eq. (1) to the Hamiltonian of the ferromagnetic Ising model, which again has a transition at $T_M = 1$. (Note the difference in how the strength of the coupling scales with $N$.) The ground states of both the annealed and the Mattis models are unfrustrated, but the annealed case has extensive degeneracy. In fact, any spin configuration can be made a ground state through a suitable choice of the $J_{ij}$'s.

The quenched SK model can be extended to an uneven distribution of ferromagnetic and antiferromagnetic bonds. Let $p$ be the percentage of bonds with $J_{ij} = J > 0$, and $1 - p$ the percentage of bonds with $J_{ij} = -J$. Nishimori discovered a special line on the $p$-$T$ plane where configuration-dependent quantities, such as the internal energy, acquire the same value when averaged over the equilibrium distributions in the quenched ensemble, or obtained from the annealed model at the same temperature [19, 20]. In the present case, the line is given by

$$e^{2J/T} = \frac{p}{1-p}. \tag{4}$$

When $p$ is significantly larger than $1/2$, we expect the ground state of the quenched model to be ferromagnetic. An estimate for the stability boundary of this phase can be obtained from the following scaling argument. The mean energy for a single spin is given by $e_f \simeq (2p-1)NJ$, and its standard deviation $\delta e_f \simeq [4p(1-p)]^{1/2}N^{1/2}J$. Setting $e_f = \delta e_f$ yields a critical value $p_c$ at the onset of the fluctuation-dominated spin-glass phase. In terms of the scaled variable $r = (2p-1)N^{1/2}$, we have $r_c \simeq 1$. The temperature scale for thermal-induced transition is set by $e_f = r_c N^{1/2}J$. Under our convention $J = N^{-1/2}$, the critical temperature $T_c$ for the spin-glass transition is of order 1.

Combining with Eq. (4), we show in Fig. 1 the generic phase diagram for the extended SK model, where the spin glass transition expands to a line from P to N [20, 21]. As temperature decreases, the annealed SK model transits from the paramagnetic phase to the "ferromagnetic" or Mattis phase along the Nishimori line, passing through the multicritical point N. The nature of this transition is our main focus here.

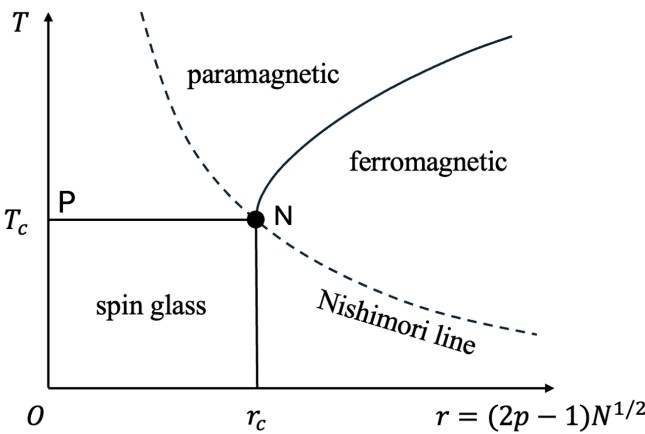

Figure 1: Schematic phase diagram of the extended SK model. The spin glass phase has a finite width in the scaled variable $r$. The Nishimori point N is a multicritical point signifying the onset of Mattis order in the annealed model.

## 2.2 Random matrix formulation and spectral representation

Given the quadratic form of Eq. (1), it is tempting to rewrite the Hamiltonian in normal coordinates $s_k = \vec{S} \cdot \vec{V}_k$,

$$H = -\frac{1}{2}\vec{S}\mathbb{J}\vec{S}^T = -\frac{1}{2}\sum_k^N \lambda_k s_k^2. \tag{5}$$

Here $\vec{S} = (S_1, S_2, ..., S_N)$ represents the spin state, and $\vec{V}_k, k = 1, \ldots, N$, are the normalized eigenvectors of the $N \times N$ dimensional matrix $\mathbb{J} = \{J_{ij}\}$, with corresponding eigenvalues $\lambda_1 < \lambda_2 < \ldots < \lambda_N$ arranged in ascending order. For the sake of convenience, we refer to $\vec{V}_N$, the eigenvector associated with the largest eigenvalue $\lambda_N$, as the principal eigenvector, and $s_N$ the principal eigenmode.

Under the spectral decomposition (5), summation over the spins in Eqs. (2) and (3) can be readily carried out, leaving behind an effective energy function governing the statistics of the eigenvalues.[1] Computation of the phase diagram shown in Fig. 1 is then reduced to the analysis of the eigenvalue spectrum that enables various universal features and exact results to be derived thanks to the extensive literature on the random matrix theory. For completeness, we sketch the main steps of this approach below and refer the reader to Refs. [1, 22] for additional technical details.

The real symmetric matrix $\mathbb{J} = \{J_{ij}\}$ of independently distributed coupling constants forms the well-known Gaussian-orthogonal ensemble (GOE). Their eigenvalues are distributed according to

$$P_q(\vec{\lambda}) \sim \exp[-Nf_q(\vec{\lambda})], \tag{6}$$

where $\vec{\lambda} = (\lambda_1, \lambda_2, \ldots, \lambda_N)$ and

$$f_q(\vec{\lambda}) = \frac{1}{2}\sum_{k=1}^N V_0(\lambda_k) - \frac{1}{N}\sum_{1 \le j < k \le N} \ln|\lambda_k - \lambda_j|. \tag{7}$$

Here the potential $V_0(\lambda)$ is related to the bare distribution of the $J_{ij}$'s, e.g., whether it is continuous or taking only discrete values. However, when the central limit theorem applies, we may write $V_0(\lambda) = \lambda^2/(2N\sigma^2)$ [23], where $\sigma$ is the standard deviation of the $J_{ij}$'s which is equal to $1/\sqrt{N}$ in our case.

In the annealed SK model, collective spin fluctuations modify the distribution of the eigenvalues. Previous studies [1] and our own numerical simulations discussed in Sec.4 show that the discreteness of Ising spins does not play an important role in determining $P_a(\vec{\lambda})$ when $N$ is sufficiently large. Moving to the spherical model of continuous spins under the constraint $\sum_{k=1}^N s_k^2 = N$, we may write,

$$P_a(\vec{\lambda}) = A\int[d\vec{s}]P_q(\vec{\lambda})e^{\frac{\beta}{2}\sum_k \lambda_k s_k^2}\delta\left(\sum_k s_k^2 - N\right), \tag{8}$$

where $A$ is a normalization constant. Integration over the spin components can be readily carried out with the help of the identity

$$\delta(x) = \frac{1}{2\pi i}\int_{z_0-i\infty}^{z_0+i\infty} e^{zx}dz, \tag{9}$$

---

[1]Strictly speaking, this procedure applies only when there is no replica symmetry breaking. In the quenched SK model, for example, discrete Ising spins give rise to a complex energy landscape in the low temperature phase that needs to be treated differently. See, e.g., Ref. [22].

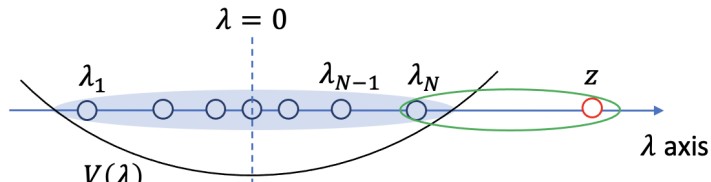

Figure 2: One-dimensional Coulomb gas of $N$ identical charges in a confining potential $V(\lambda)$ and an extra half-charge of opposite sign at the right end.

where $z_0$ is an arbitrary real number. We further make the substitution $z \to \beta z/2$ and obtain,

$$P_{\mathrm{a}}(\vec{\lambda}) = \text{const} \times \int dz \exp[-N f_{\mathrm{a}}(\vec{\lambda}, z)], \tag{10}$$

where

$$f_{\mathrm{a}}(\vec{\lambda}, z) = f_{\mathrm{q}}(\vec{\lambda}) + \frac{1}{2N} \sum_{k=1}^{N} \ln(z - \lambda_k) - \frac{\beta}{2} z. \tag{11}$$

The parameter $z$ in Eq. (11) needs to be greater than the largest eigenvalue $\lambda_N$.

A physical interpretation of the expression (11) was given, i.e. the energy of a linear array of $N$ negative charges located at the $\lambda_k$'s and a positive charge at $z$, interacting with a logarithmic potential. Furthermore, the negative charges experience a harmonic confining potential while the positive charge, confined to the right of all the negative charges, experiences a linear potential whose strength increases with $\beta$.

## 2.3 Condensate formation at low temperatures

For a given coupling matrix $\mathbb{J}$ and a set of eigenvalues $\{\lambda_k\}$, equipartition theorem under the Hamiltonian (5) yields the thermal average

$$\langle s_k^2 \rangle = \frac{T}{z - \lambda_k}, \tag{12}$$

where $z$ (the saddle point solution) is chosen to satisfy the spherical constraint

$$\sum_{k=1}^{N} \langle s_k^2 \rangle = \sum_{k=1}^{N} \frac{T}{z - \lambda_k} = N. \tag{13}$$

As the temperature decreases, $z$ moves closer to the spectral edge $\lambda_N$. Below a critical temperature $T_c$, $\langle s_N^2 \rangle$ grows to a magnitude of order $N$, i.e., condensation of spin configurations onto the principal eigenvector takes place.

Previous work has established that $T_c = 1$ for both quenched spherical [21] and annealed [1,3,4] SK models. In the low-temperature phase, the amplitude of spin fluctuations onto the principal eigenvector is predicted to be,

$$\langle s_N^2 \rangle = \begin{cases} N(1-T), & \text{quenched,} \\ N(1-T^2), & \text{annealed} \end{cases} \qquad \text{(low-temperature phase).} \tag{14}$$

Thus $z - \lambda_N$ needs to be of order $N^{-1}$ so as to be consistent with Eq. (12). This is much smaller than the typical size of the gap $\lambda_N - \lambda_{N-1}$ which is of order $N^{-2/3}$ [12] or greater. To resolve the condensation process in the critical region, one needs to examine carefully granularity of the spectrum at the spectral edge.

## 3 The maximum likelihood spectrum

### 3.1 Stieltjes transform and solution at infinite size

In the limit of large $N$, the probability distributions Eqs. (6) and (10) are sharply peaked at their respective maximum likelihood eigenvalues that satisfy

$$\frac{\partial f_{q,a}(\vec{\lambda})}{\partial \lambda_k} = \frac{1}{2}V'(\lambda_k) - \frac{1}{N}\sum_{j\neq k}\frac{1}{\lambda_k - \lambda_j} = 0 \qquad (k=1,\ldots,N). \tag{15}$$

Here $V(\lambda) = V_0(\lambda)$ in the quenched case and $V(\lambda) = V_0(\lambda) + N^{-1}\ln(z-\lambda)$ in the annealed case.

To explore analytic properties of the maximum likelihood spectrum, it is customary to introduce Stieltjes transform

$$g_N(x) = \frac{1}{N}\sum_{k=1}^{N}\frac{1}{x-\lambda_k}, \tag{16}$$

where the $\lambda_k$'s are simple poles of $g_N(x)$ on the complex plane. Under Eq. (15), $g_N(x)$ obeys the nonlinear first order ordinary differential equation [23],

$$V'(x)g_N(x) - \Pi_N(x) = g_N^2(x) + \frac{1}{N}g_N'(x), \tag{17}$$

with

$$\Pi_N(x) \equiv \frac{1}{N}\sum_{j=1}^{N}\frac{V'(x)-V'(\lambda_j)}{x-\lambda_j}. \tag{18}$$

For $V_0(\lambda) = \lambda^2/2$, simple algebra gives

$$\Pi_N(x) = \begin{cases} 1, & \text{quenched,} \\ 1 + N^{-1}g_N(z)/(x-z), & \text{annealed.} \end{cases} \tag{19}$$

The parameter $z$ satisfies Eq. (13) which now takes the form

$$g_N(z) = \beta. \tag{20}$$

In the limit $N \to \infty$, Eq. (17) reduces to a quadratic equation for $g_\infty(x)$ whose solution is given by

$$g_\infty(x) = \frac{x - \sqrt{x^2 - 4}}{2}. \tag{21}$$

As required by Eq. (16), $g_\infty(x)$ decreases monotonically with $x$ on the $x > 2$ side, starting from $g_\infty(2) = 1$. The corresponding spectral density of eigenvalues satisfies the well-known semi-circle law [23],

$$\rho(\lambda) = \frac{\sqrt{4-\lambda^2}}{2\pi}. \tag{22}$$

As discussed in detail in Ref. [1], Eqs. (20) and (21) are compatible with each other only when $\beta < 1$ or $T > 1$. In this regime, the two equations combine to give

$$z(\beta) = \beta + \beta^{-1}. \tag{23}$$

For $\beta > 1$, the principal eigenvalue $\lambda_N$ merges with $z$ such that its contribution to $g_N(z)$ becomes of order 1 and needs to be isolated. Excluding the term $k = N$ from the sum in (16), we rewrite Eq. (20) as,

$$g_{N-1}(z) + \frac{1}{N}\frac{1}{z-\lambda_N} = \beta. \tag{24}$$

In the quenched case, replacing $g_{N-1}(z)$ with $g_\infty(z)$ yields

$$z_q(\beta) = \lambda_N + \frac{1}{N(\beta - 1)} \qquad (\beta > 1). \tag{25}$$

In the annealed case, $z$ and $\lambda_N$ are obtained by solving Eq. (24) together with Eq. (15) at $k = N$. The latter is given by,

$$\frac{\lambda_N}{2} - \frac{1}{2N} \frac{1}{z_a - \lambda_N} - g_{N-1}(\lambda_N) = 0. \tag{26}$$

In the large $N$ limit, we have $\lambda_N \simeq z_a(\beta) = \beta + \beta^{-1}$ with

$$z_a(\beta) - \lambda_N = \beta/[N(\beta^2 - 1)] \qquad (\beta > 1). \tag{27}$$

Equations (25) and (27) give the results in Eq. (14).

## 3.2 Solution at large but finite size

### 3.2.1 Scaling analysis at the spectral edge

To gain a better understanding of the spectral evolution across the transition at $T_c = 1$, we examine Eq. (17) more systematically, treating $\epsilon \equiv 1/N$ as a small parameter. This problem was considered previously in Ref. [9] under an approximate scheme.

Specializing on $V_0(\lambda) = \lambda^2/2$, we write Eq. (17) explicitly in the form,

$$g_N^2(x) - x g_N(x) + 1 = -\epsilon g_N'(x) + \epsilon \frac{g_N(x) - g_N(z)}{x - z} \qquad \text{(annealed model).} \tag{28}$$

The quenched case is recovered by dropping the last term on the right-hand-side of Eq. (28).

Starting from the zeroth order solution (21) and focusing on the edge of the spectrum at $x \simeq 2$, we introduce the scaling ansatz

$$g_N(x) = \frac{x}{2} - \epsilon^{1/3} \varphi\big((x - 2)\epsilon^{-2/3}\big). \tag{29}$$

The expanded spectral axis on the scale of the gap between leading eigenvalues allows one to resolve individual poles of $g_N(x)$. Substituting (29) into Eq. (28) and keeping terms up to order $\epsilon^{2/3}$, we obtain,

$$\varphi'(u) = \varphi^2(u) - u + \frac{\varphi(u) - \varphi(c)}{u - c} \qquad \text{(annealed model).} \tag{30}$$

Here $u = (x - 2)N^{2/3}$ and $c = (z - 2)N^{2/3}$ are the scaled variables. To be consistent with Eq. (21), we demand

$$\varphi(u) \simeq u^{1/2}, \tag{31}$$

at large and positive $u$.

At a given temperature $T = 1/\beta$, we introduce a scaled variable

$$\Delta \equiv \frac{1}{2} N^{1/3}\Big(\frac{1}{\beta} - \beta\Big), \tag{32}$$

to measure deviations from the critical point at $\Delta = 0$. Note that $(\beta^{-1} - \beta)/2 \simeq T - T_c$. Equation (20) with (23) yields, to the leading order in $\epsilon$,

$$\varphi(c) = \Delta. \tag{33}$$

Apart from the nonlocal term on the right-hand-side, Eq. (30) is known as a first order ordinary differential equation of the Riccati type. The substitution

$$\varphi(u) = -\frac{Q'(u)}{Q(u)}, \tag{34}$$

brings it to a linear second order equation,

$$Q''(u) = uQ(u) + \frac{Q'(u) + \Delta Q(u)}{u - c}, \tag{35}$$

with the condition $Q'(c) = -\Delta Q(c)$. The asymptotic form (31) sets the boundary condition

$$\lim_{u \to \infty} Q(u) = 0. \tag{36}$$

The transformation (34) converts poles of $\varphi(u)$ at $u_1 > u_2 > \dots$ (starting from the largest one on the right) into nodes of $Q(u)$, where $Q(u_k) = 0$. The eigenvalues at the leading edge are then obtained, to the leading order in $\epsilon$,

$$\lambda_{N-k+1} = 2 + u_k N^{-2/3} \qquad (k = 1, 2, \dots). \tag{37}$$

### 3.2.2 Largest eigenvalues in the quenched model

For the quenched model, spectrum of the coupling matrix is not affected by spin fluctuations. The equation governing $Q(u)$ can be obtained from (35) by taking the limit $c \to \infty$, i.e.,

$$Q''(u) = uQ(u) \qquad \text{(quenched)}. \tag{38}$$

Solutions to Eq. (38) can be expressed as a linear superposition of the Airy functions $\text{Ai}(u)$ and $\text{Bi}(u)$, whose asymptotic behavior at $u \gg 1$ are given by $\exp(-2u^{3/2}/3)$ and $\exp(2u^{3/2}/3)$, respectively. The requirement (36) selects $Q(u) = \text{Ai}(u)$.

On the $u < 0$ side, Eq. (38) resembles that of a harmonic oscillator but with a spring constant that grows with $|u|$. This observation suggests approximate solutions of the form

$$Q(u) \simeq A \sin(|u|^{3/2}) + B \cos(|u|^{3/2}). \tag{39}$$

Therefore the nodes are approximately located at

$$u_k \simeq u_0 - (k\pi)^{2/3} \qquad (k = 1, 2, \dots), \tag{40}$$

where $u_0$ is an offset. The density of nodes thus increases as $|u_k|^{1/2}$, in agreement with the square-root singularity of Eq. (22) at the spectral edge. We note in passing that the close connection between the Coulomb gas problem and nodes of the Airy function has also appeared in previous work in somewhat different contexts [7, 24].

### 3.2.3 Spectral crossover in the annealed model

We now consider the more general case of Eq. (35), where the point $c$ needs to be determined self-consistently at a given $\Delta$. Far away from the origin, the last term on the right-hand-side of the equation is smaller than the other two terms. Consequently, the asymptotic behavior of the solution is the same as that of $\text{Ai}(u)$.

The spurious singular behavior of Eq. (35) at $u = c$ can be dealt with by considering a series solution in $v = u - c$,

$$Q(c + v) = \sum_{n=0}^{\infty} a_n v^n. \tag{41}$$

Table 1: First five nodes $u_k^0$ and $u_k^c$ of Ai$(u)$ and $Q(u)$ at the critical point $\Delta = 0$, respectively, computed to eight significant digits using the series expansion.

| $k$ | 1 | 2 | 3 | 4 | 5 |
|---|---|---|---|---|---|
| $u_k^0$ | $-2.3381074$ | $-4.0879494$ | $-5.5205598$ | $-6.7867080$ | $-7.9441336$ |
| $u_k^c$ | $-1.0187930$ | $-3.2481977$ | $-4.8200993$ | $-6.1633075$ | $-7.3721773$ |

Substituting into Eq. (35), we obtain

$$a_1 = -\Delta a_0\,, \tag{42}$$

$$a_n = \frac{1}{n(n-2)}(\Delta a_{n-1} + \Delta^2 a_{n-2} + a_{n-3}) \qquad (n \geq 3)\,. \tag{43}$$

At $n = 2$, terms containing $a_2$ cancel and one is left with

$$c = \Delta^2 \qquad \text{(annealed model)}. \tag{44}$$

The coefficient $c_2$ needs to be chosen to match the asymptotic behavior $Q(u) \sim \exp(-2u^{3/2}/3)$ at large positive $u$.

We have computed the function $Q(u)$ from the series solution. Nodes $\{u_k\}$ can also be computed using a simple algorithm such as the bisection method. Figure 3 shows our results for three different values of $\Delta$. Also shown is the Airy function Ai$(u)$ for the quenched model (shaded curve). In general, the $k$th node $u_k(\Delta)$ increases monotonically as $\Delta$ decreases, with $u_k(\infty) = u_k^0$ and $u_k(-\infty) = u_{k-1}^0 > u_k^0$. Here $\{u_k^0\}$ are nodes of Ai$(u)$, with values of the first five nodes given in Table 1. As a convention, we set $u_0^0 = +\infty$.

Figure 4 shows the first (black) and second (red) gap at the edge of the spectrum as well as $z - \lambda_N = [\Delta^2 - u_1(\Delta)]N^{-2/3}$ (blue) against $\Delta$ across the critical region. On the high-temperature side ($\Delta > 0$), the two gaps approach their respective asymptotic values that can be computed from Table 1). On the low-temperature side, $u_1(\Delta) \simeq c = \Delta^2$ and $u_2(\Delta) \to u_1^0$, so that the first gap follows the diverging curve $\Delta^2 - u_1^0$ indicated by the dashed line. The latter also gives the asymptotic behavior of $(z - \lambda_N)N^{2/3}$ on the high-temperature side, in

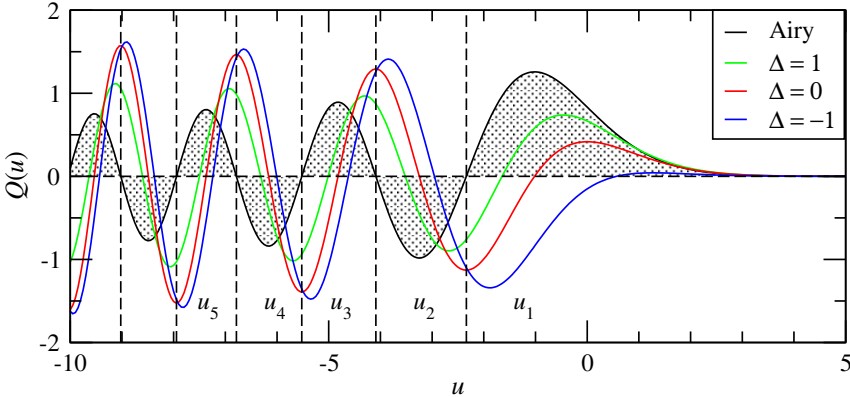

Figure 3: Edge of the maximum likelihood spectrum with the largest eigenvalues $\lambda_{N-k+1} \simeq 2 + u_k N^{-2/3}$. As the parameter $\Delta$ decreases across the critical point $\Delta = 0$, the nodes $u_k$ of $Q(u)$ move to the right within each of the intervals defined by successive nodes of the Airy function Ai$(u)$ (vertical dashed lines). The first node $u_1$ moves to infinity as $\Delta$ decreases further into the low-temperature phase.

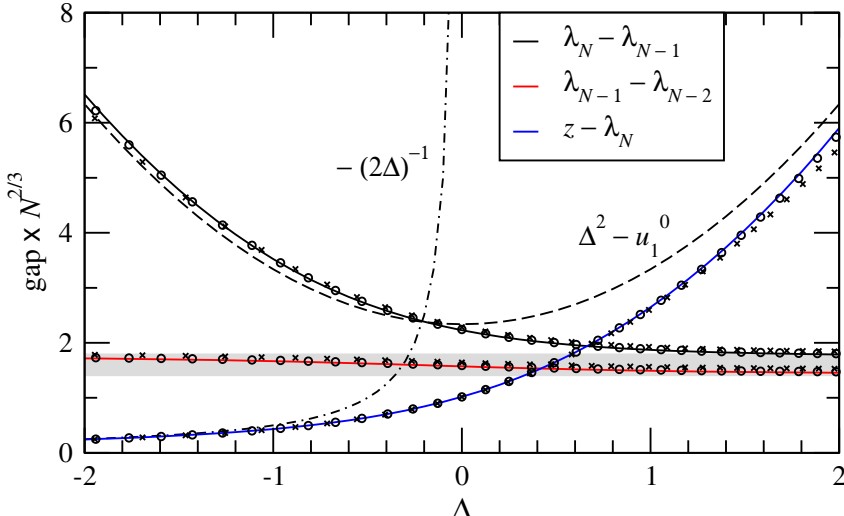

Figure 4: Gap scaling in the critical region showing crossovers in a system of large but finite size. On the low-temperature side, the principal eigenvalue $\lambda_N$ separates from the rest of the spectrum. Also shown are the numerical solutions of Eqs. (15) together with Eq. (23) for $N = 32$ (crosses) and $N = 256$ (circles).

agreement with the data shown by the blue curve. The low-temperature side of the blue curve approaches the function $-(2\Delta)^{-1}$ predicted by Eq. (27) (dash-dotted line). Together they provide a detailed description of the spectral crossover in the transition region for a large but finite system.

As a check of the analysis based on Stieltjes transform, we also performed numerical solution of Eqs. (15) together with Eq. (23) for systems sizes up to $N = 512$. Our results for the scaled gaps at two selected sizes are included in Fig. (4) for comparison. It is clear that corrections to the leading order results are already quite small at $N = 32$ and decrease further as $N$ increases.

## 4 Finite-size scaling of collective spin components

### 4.1 Maximum likelihood amplitudes

With the maximum likelihood spectrum obtained in Sec. 3.2, we are ready to establish finite-size scaling expressions for the spin components $\langle s_k^2 \rangle$ across the transition under Eq. (12).

#### 4.1.1 Spin condensation in the quenched model

In the quenched model, the eigenvalue spectrum does not change with temperature. To predict $\langle s_k^2 \rangle$ in the critical region, we need to determine $z$ from Eq. (13). In terms of the scaled variables $\Delta = N^{1/3}(T - T^{-1})/2$ and $c = (z - 2)N^{2/3}$, Eq. (33) now takes the form,

$$\Delta = -\frac{\text{Ai}'(c)}{\text{Ai}(c)} \qquad \text{(quenched model)}. \qquad (45)$$

Figure 5 shows $c$ against $\Delta$ (blue line) obtained from numerical evaluation of the right-hand-side of Eq. (45). Also shown is Eq. (44) for the annealed case (red line). The two curves approach each other on the high-temperature side. On the low-temperature side, $c_q(\Delta)$ for the quenched model approaches the first node of $\text{Ai}(u)$ at $u_1^0$. Noting that this corresponds to

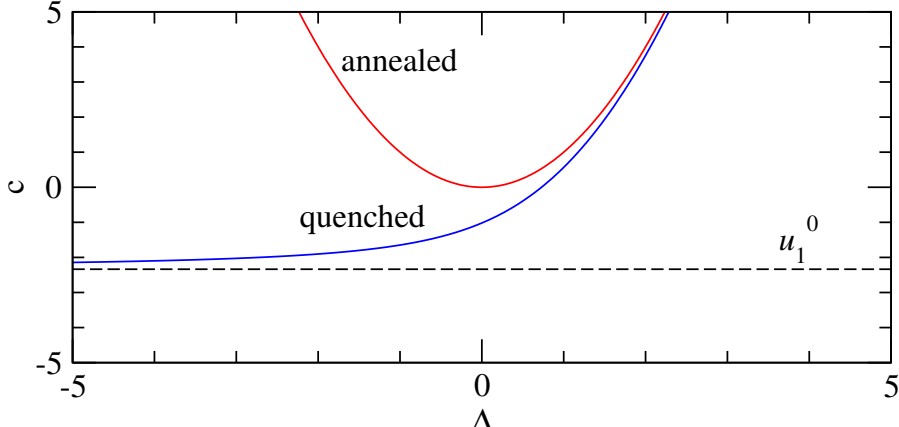

Figure 5: Relation between $c$ and $\Delta$ for quenched (blue) and annealed (red) models. The dashed line gives the minimal value for $c$. A larger $c$ suppresses amplitudes of spin fluctuations apart from the principal component.

a first order node of the Airy function, we have

$$c_q(\Delta) \simeq u_1^0 - \Delta^{-1} \qquad (\Delta \ll -1). \tag{46}$$

With $c_q(\Delta)$ given above, we obtain spin components in the critical region in scaling form,

$$\langle s_{N-k+1}^2 \rangle = \frac{TN^{2/3}}{c_q(\Delta) - u_k^0} \equiv TN^{2/3}\psi_k^q\left(N^{1/3}\frac{T^2-1}{2T}\right) \qquad (k = 1, 2, \ldots). \tag{47}$$

Here $\psi_k^q(u) = [c_q(u) - u_k^0]^{-1}$ are scaling functions associated with the leading modes.

In the low-temperature phase, substituting Eq. (46) into Eq. (47), we obtain

$$\begin{aligned} \langle s_N^2 \rangle &= \frac{T}{z - \lambda_N} = N(1 - T), \\ \langle s_{N-k+1}^2 \rangle &= \frac{TN^{2/3}}{u_1^0 - u_k^0} \qquad (k = 2, 3, \ldots) \end{aligned} \qquad (T < T_c). \tag{48}$$

Equation (14) for the quenched spherical model is reproduced.

Far away from the critical region on the high-temperature side, $c_q(\Delta) \simeq \Delta^2 \gg u_k^0$. Equation (47) then reduces to

$$\langle s_{N-k+1}^2 \rangle \simeq \frac{T}{z(\beta) - 2} = \frac{T^2}{(T-1)^2} \qquad (T > T_c), \tag{49}$$

for the leading modes, where Eq. (23) is used.

### 4.1.2 Spin condensation in the annealed model

The scaling behavior of $\langle s_k^2 \rangle$ in the annealed model can be established in a similar way as in the quenched case. Quite generally, $c = \Delta^2$ and hence the maximum likelihood estimation (MLE) of the spectrum yields

$$\langle s_{N-k+1}^2 \rangle_{\mathrm{MLE}} = \frac{TN^{2/3}}{\Delta^2 - u_k(\Delta)} \equiv TN^{2/3}\psi_k^a\left(N^{1/3}\frac{T^2-1}{2T}\right) \qquad (k = 1, 2, \ldots). \tag{50}$$

The scaling functions $\psi_k^a(u)$ differ from $\psi_k^q(u)$ in two ways. One is the relation between $c$ and $\Delta$ as illustrated in Fig. 5. The second is that $u_k(\Delta)$ approaches $u_k^0$ only when $\Delta$ becomes large and positive (see Fig. 3). Therefore spin fluctuations become nearly identical in the two cases only in the high-temperature phase.

Outside the critical region and into the low-temperature phase, Eqs. (23) and (27) together give,

$$\langle s_N^2 \rangle = \frac{T}{z - \lambda_N} = N(1 - T^2),$$
$$\langle s_{N-k+1}^2 \rangle = \frac{T}{z - \lambda_{N-k+1}} \simeq \frac{T^2}{(1-T)^2} \qquad (k = 2, 3, \ldots) \qquad (T < T_c). \qquad (51)$$

Again we recover the annealed result in Eq. (14). Note that fluctuation amplitudes in the annealed model are much weaker when compared to the quenched spherical model, except for the principal component.

## 4.2 Monte Carlo simulations of the annealed SK model

To gain a quantitative understanding of fluctuation effects not accounted for by the maximum likelihood spectrum, we carried out Monte Carlo (MC) simulations of the annealed SK model. In our implementation of the Metropolis algorithm for both spin and $J_{ij}$ updates, a MC sweep of all $N$ spins is performed following an attempted update of a randomly chosen bond. We refer to this as one MC step in our simulation. The energy involved in a single bond flip is $\Delta E_J = \pm 2/\sqrt{N}$, hence the acceptance rate is quite high except at very low temperatures. The energy change $\Delta E_S$ for a spin flip, on the other hand, is of order 1.

In a typical simulation at a given temperature $T$, we equilibrate an initially random configuration of spins and bonds for $5N^2$ MC steps, followed by $20N^2$ MC steps for data collection. This is found to be sufficient for the computation of equilibrium properties of systems up to $N = 512$.

We performed 200 independent simulations for each system size and temperature. Data collected included spin configurations and the coupling matrices, sampled every $N$ bond flip attempts to determine equilibrium properties. Additionally, we tested varying the number of spin flips between successive bond flip attempts, from $N/16$ to $4N$, and found consistent results across different setups.

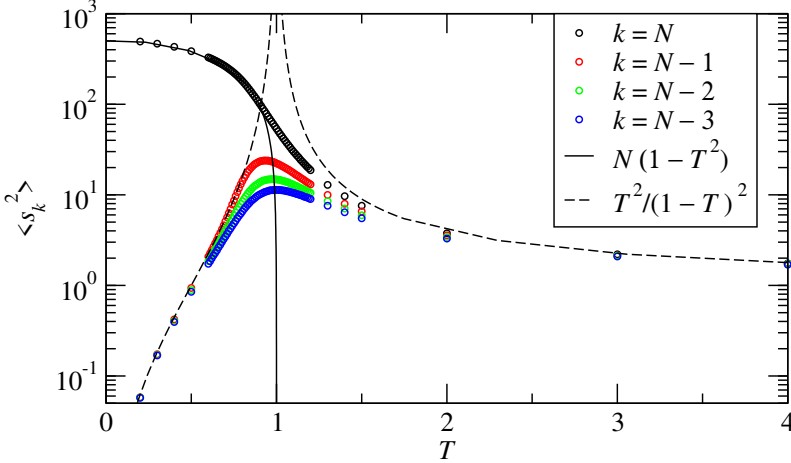

Figure 6: Mean-square amplitudes of the first four spin components against temperature $T$ from simulations of the $N = 512$ annealed SK model. Onset of condensation is seen from the rise of the principal component below $T = 1$.

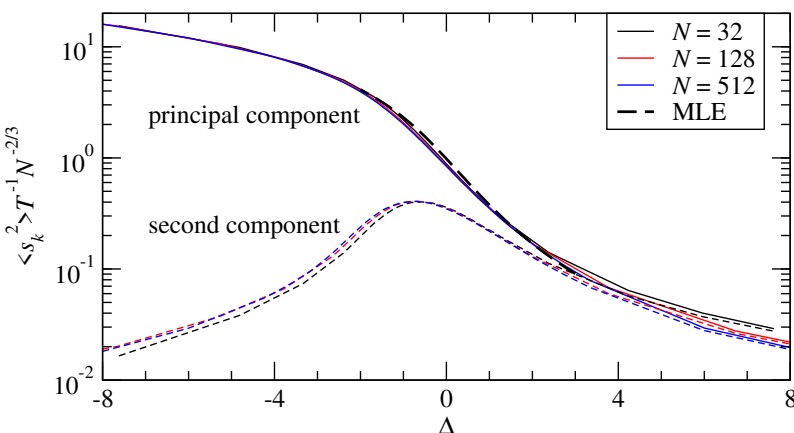

Figure 7: Finite-size scaling of the mean-square amplitude of the first ($k = N$) and second ($k = N - 1$) component of spin fluctuations across the transition. The difference between the maximum likelihood estimation (MLE) and the full annealed SK model becomes more pronounced at the transition point $\Delta = 0$.

Figure 6 presents mean-square fluctuations of the first four components of spin configurations obtained from our simulations. To compute the spin components, we projected the spins onto the eigenvectors of the instantaneous coupling matrix. Away from the critical region, the data agree well with the predicted behavior given by Eqs. (49) and (51) (dashed and solid lines).

To check the finite-size scaling predictions, we plot in Fig. 7 the scaled mean-square amplitudes of the principal and second component against $\Delta$ for three different sizes. Indeed, data at different system sizes collapse well in the whole temperature range. The dashed line in the figure is obtained by replacing the average over the eigenvalue spectrum with the maximum likelihood values obtained in Sec. 3. Their difference is more evident near the transition point $\Delta = 0$.

Figure 8 shows the ratio between the mean-square amplitudes and their MLE values for the first four components of spin fluctuations at the critical point. In agreement with Fig. 7, fluctuations of the principal component is weaker than predicted based on the MLE. On the other hand, the second, third and fourth component have stronger fluctuations in the full model as compared to the MLE predictions. Further work is needed to understand these observations in terms of collective fluctuations of the eigenvalue spectrum.

## 5 Summary and conclusions

This paper build on recent observations made by Foini and Kurchan [1] on the fascinating properties of annealed models. We introduce a perturbative scheme to solve the exact equation satisfied by the Stieltjes transform of the maximum likelihood spectrum of the coupling matrix at finite $N$. The discrete set of eigenvalues at the leading edge of the spectrum can then be resolved exactly to order $N^{-2/3}$ in the high-temperature phase and across the transition region. Although the scaling of the edge spectral gaps against $N$, including aspects of their fluctuation behavior, has been extensively reported in the literature (see., e.g., [8, 25, 26]), the method we propose here uses only elementary algebra and goes beyond equilibrium solution of the usual Coulomb gas problem where the nodal points of the Airy function appear [7, 24].

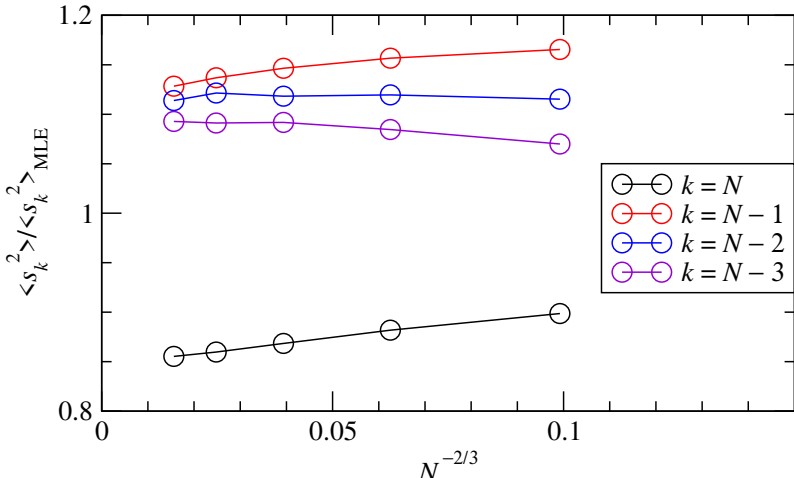

Figure 8: Ratio of spin fluctuation amplitudes at $T = T_c$ (or equivalently, $\Delta = 0$) against values predicted by the maximum likelihood estimation (MLE). Spectral fluctuations appear to reduce the amplitude of the principal component while enhance all others.

With this technique, we analyzed the spectral crossover of the annealed SK spin glass model, which exhibits various singular behaviors at the transition temperature $T_c$. Finite-size scaling forms of principal spin fluctuations were obtained in the critical region, connecting smoothly to the known results in the low and high-temperature phases. These predictions, based on the maximum likelihood spectrum, were then compared with Monte Carlo simulation results. While the scaling exponents are borne out by the simulation data, discrepancies in the scaling amplitudes were observed in the critical region. These are presumably due to fluctuations in the eigenvalue spectrum in the annealed model, which deserve further study.

The finite-size scaling results of the edge spectrum apply equally well to the spherical model with continuous spins and a matrix of quenched coupling constants. For the SK model with Ising spins, the principal component analysis fails to capture replica symmetry breaking in the low temperature spin glass phase. Nevertheless, we expect finite-size scaling properties of the SK model at criticality to be similar to those of the spherical model, making them amenable to the analysis presented in this paper.

# Acknowledgments

We would like to thank Leticia Cugliandolo and Jorge Kruchan for insightful discussions. This work is supported by Center for Computational Science and Engineering at Southern University of Science and Technology.

**Author contributions**   L-H T. designed the project and performed most of the analytic calculations. D. W. carried out numerical solution of the maximum likelihood spectrum and performed Monte Carlo simulations of the annealed SK model. Both participated in writing up the work.

**Funding information**   The work is supported in part by the Research Grants Council of the HKSAR under grants 12304020 and 12301723.

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
