# Peer review of "Exact finite-size scaling of the maximum likelihood spectra in the quenched and annealed Sherrington-Kirkpatrick spin glass"

_SciPost Physics, doi:SciPost Phys. 17, 106 (2024)_

## Round 1 · Referee Report · Anonymous (Referee 1) · 2024-7-14

Report

See attached

Attachment

Recommendation

Ask for minor revision

  • validity: -
  • significance: -
  • originality: -
  • clarity: -
  • formatting: -
  • grammar: -

Author:  Ding Wang  on 2024-08-09  [id 4684]

(in reply to Report 1 on 2024-07-14)
Category:
answer to question

We thank Reviewer 1 for the positive recommendation of our manuscript. The comments on previous related work are extremely valuable. We have studied these papers and referenced them in the revised manuscript, as explained in the itemized reply below.

**Reviewer 1:**

Still, I believe a few references to earlier literature should be added to the text, to put it in appropriate context. In particular, I can not fully agree with the statement in the end of the paper “The discrete set of eigenvalues at the leading edge of the spectrum can then be resolved exactly to order N^(-2/3), which appears to have been an open problem before the reported work”. Not only a lot is known about the mean positions, but also fluctuations of the individual eigenvalues (in particular, the whole gap distribution between two largest/smallest eigenvalues) has been addressed for a related case of GUE matrices, see e.g.

Anthony Perret, Gregory Schehr, “Near-Extreme Eigenvalues and the First Gap of Hermitian Random Matrices,” J Stat Phys (2014) 156:843–876

and also a lot is known about extreme GOE eigenvalues.

**Reply**

We have revised our statement in the concluding section and added additional references as follows:

“Although the scaling of the edge spectral gaps against N, including aspects of their fluctuation behavior, has been extensively reported in the literature (see., e.g., [8], [24], [25]), the method we propose here uses only elementary algebra and goes beyond equilibrium solution of the usual Coulomb gas problem where the nodal points of the Airy function appear [7, 23].”

[7] L. A. Pastur and M. Shcherbina, Eigenvalue distribution of large random matrices, 171. American Mathematical Soc. (2011).
[8] S. N. Majumdar and G. Schehr, Top eigenvalue of a random matrix: large deviations and third order phase transition, Journal of Statistical Mechanics: Theory and Experiment 2014(1), P01012 (2014).
[23] Y. Chen and M. E. H. Ismail, Ladder operators and differential equations for orthogonal polynomials, Journal of Physics A: Mathematical and General 30(22), 7817 (1997).
[24] A. Perret and G. Schehr, Near-extreme eigenvalues and the first gap of hermitian random matrices, Journal of Statistical Physics 156, 843 (2014).
[25] S. N. Majumdar, A. Pal and G. Schehr, Extreme value statistics of correlated random variables: a pedagogical review, Physics Reports 840, 1 (2020).

**Reviewer 1:**

In particular, appearance of the Airy equation (see eq. (38) for Q(u)) in describing properties of the eigenvalues at the edge of GOE is a classical result, see e.g. section 6.3 of the monograph

L. Pastur, M. Shcherbina “Eigenvalue Distribution of Large Random Matrices” (AMS, 2011))

**Reply**

We have included the following sentence at the end of section 3.2.2:

“We note in passing that the close connection between the Coulomb gas problem and nodes of the Airy function has also appeared in previous work in somewhat different contexts [7, 23].”

See also our reply in the previous page.

**Reviewer 1:**

What concerns the positions of resolvent poles/zeros of Q(u) given in eqs. (37)-(40), it reminded me the study of positions of extreme zeroes of Hermite polynomials, described in the Theorem 5.2 in p.7826 of the paper

Yang Chen and Mourad E H Ismail
“Ladder operators and differential equations for orthogonal polynomials,” J. Phys. A: Math. Gen. 30 (1997) 7817–7829

**Reply**

This is an interesting article, though somewhat outside the scope of present paper. We have made reference to this work (Ref. [23]) at the end of Sec. 3.2.2 as well as in the concluding section.

**Reviewer 1:**

Closer to the topic of the present paper, the scaling analysis at the spectral edge, section 3.2.1, looks to me overall not dissimilar to the analysis in the section IV of the paper

Finite-size critical scaling in Ising spin glasses in the mean-field regime
T. Aspelmeier, H. G. Katzgraber, D. Larson, M. A. Moore, Matthew Wittmann, and J Yeo Phys. Rev. E 93, 032123 (2016)

which I feel also deserves mentioning in the present context.

**Reply**

This is indeed a very interesting paper that develops a finite-size analysis based on the spin-glass susceptibility χ_l. Their treatment of the edge spectrum captures the correct physics behind scaling properties of χ_l near the glass transition. Nevertheless, Eq. (28) in their paper is only approximate and does not follow from a systematic expansion such as the one presented in this paper. In the revised version, we added the following sentence in the third paragraph of the introductory section:
“To achieve this, it is essential to resolve the spacing between successive eigenvalues of the coupling matrix at the spectral edge, which, surprisingly, has not been adequately addressed despite many closely related work and results (see, e.g., [7–9]).”
[9] T. Aspelmeier, H. G. Katzgraber, D. Larson, M. A. Moore, M. Wittmann and J. Yeo, Finite-size critical scaling in ising spin glasses in the mean-field regime, Phys. Rev. E 93, 032123382 (2016).
Furthermore, we added a sentence at the end of the first paragraph in Sec. 3.2.1:

“This problem was considered previously in Ref. [9] under an approximate scheme.”

**List of Changes**

In addition to those listed above, we have slightly modified the abstract to reflect the changes made in the main text.

---

## Round 1 · Referee Report · Anonymous (Referee 2) · 2024-7-29

Report

The report is attached

Attachment

Recommendation

Ask for minor revision

  • validity: -
  • significance: -
  • originality: -
  • clarity: -
  • formatting: -
  • grammar: -

Author:  Ding Wang  on 2024-09-04  [id 4735]

(in reply to Report 2 on 2024-07-29)

We thank Reviewer 2 for a careful reading of our manuscript and his/her positive recommendations. Below is our reply to the reviewer’s comment:

Distinction between Ising and Spherical Models:
While the physics in the annealed case is similar the low temperature physics of the quenched problem is different. The authors should probably be careful with this distinction in their discussion?

Reply:

This distinction is indeed very important and should be emphasized. In the revised manuscript, we added a footnote in the second paragraph of Sec. 2.2 after Eq. (5) with a new reference:

“Strictly speaking, this procedure applies only when there is no replica symmetry breaking. In the quenched SK model, for example, discrete Ising spins give rise to a complex energy landscape in the low temperature phase that needs to be treated differently. See, e.g., Ref. [22].”

[22] C. De Dominicis and I. Giardina, Random fields and spin glasses: a field theory approach, Cambridge University Press, ISBN 9780521847834 (2006).

Furthermore, the following sentence is added in Section 2.3, after Eq. (13):

“However, in the quenched Ising SK model, the discrete spins and frustrations prevent condensation below the critical temperature, leading the system to form a spin glass instead.”

Additionally, we note that although the low-temperature physics differ, the quenched Ising and spherical models exhibit the same finite-size scaling near the critical temperature according to our results and those in the literature, e.g., T. Aspelmeier, H. G. Katzgraber, D. Larson, M. A. Moore, M. Wittmann and J. Yeo, Finite-size critical scaling in Ising spin glasses in the mean-field regime, Phys. Rev. E 93, 032123 (2016). The paragraph below is included at the end of Sec. 5 to this effect:

“The finite-size scaling results of the edge spectrum apply equally well to the spherical model with continuous spins and a matrix of quenched coupling constants. For the SK model with Ising spins, the principal component analysis fails to capture replica symmetry breaking in the low temperature spin glass phase. Nevertheless, we expect finite-size scaling properties of the SK model at criticality to be similar to those of the spherical model, making them amenable to the analysis presented in this paper.”

We have made a few other minor rewordings throughout the paper to emphasize the distinction, and we thank the reviewer wholeheartedly for the suggestion.

---

## Round 1 · Referee Report · Anonymous (Referee 3) · 2024-8-25

Report

The present manuscript deals with an interesting problem: the finite-size scaling of the spectrum of eigenvalues in the SK model.
However, I see a main problem: the authors claim to have an exact formula for the finite-size spectrum as the title announces. However, when compared with Monte Carlo simulations, the analytical results differ from the numerical estimates.
So, either there are problems in how the numerical simulations are carried out, or the analytical result obtained by the authors is not exact but approximate.
This needs to be clarified before the manuscript can be published.

In case the analytical formula is not exact, the manuscript needs to be revised (starting from the title) and the hypotheses leading to an approximate result need to be stressed.

The other possibility is that Monte Carlo simulations are not measuring exactly what the authors are computing from the analytical formulas.
I believe the authors have given too few details about the numerical simulations.
For example, it is not clear whether they compute the eigenvectors of the coupling matrix J (which evolved during the numerical simulation) at each Monte Carlo step to compute the components.
More importantly, the authors are using a fixed ratio between the number of spin flips and bond flips: 1 bond flip for every N spin flips. How can they be sure this ratio would sample from the annealed measure?
I suggest the authors perform a more extensive set of numerical simulations where M bond flips are made for every N spin flips: if they observe the same results for any positive M value, then this a good evidence the annealed measure is sampled.
Remind that for M=0 one is sampling the quenched measured, so M=1 (the value used by the authors) can be too small and provide results from a sort of mixed ensemble.

If the discrepancy between the analytical formula and the numerical simulations is resolved, I am very in favor of publishing this work.

Recommendation

Ask for major revision

  • validity: -
  • significance: -
  • originality: -
  • clarity: -
  • formatting: -
  • grammar: -

Author:  Ding Wang  on 2024-09-04  [id 4736]

(in reply to Report 3 on 2024-08-25)

We appreciate the reviewer's careful reading of our manuscript and his/her valuable feedback. The questions and comments provided are pertinent and have helped us improve the clarity of the paper. Below, we provide a detailed response to the points raised.

Reviewer 3:

…the authors claim to have an exact formula for the finite-size spectrum as the title announces. However, when compared with Monte Carlo simulations, the analytical results differ from the numerical estimates.

So, either there are problems in how the numerical simulations are carried out, or the analytical result obtained by the authors is not exact but approximate.

This needs to be clarified before the manuscript can be published.

In case the analytical formula is not exact, the manuscript needs to be revised (starting
from the title) and the hypotheses leading to an approximate result need to be stressed.

The other possibility is that Monte Carlo simulations are not measuring exactly what the authors are computing from the analytical formulas.

Reply:

We apologize for any confusion caused by the word “exact” in the title of the paper, which may seem contradictory to Figs. 7 and 8. The exact finite-size scaling refers to the maximum-likelihood estimation (MLE) of the spectrum of the coupling constant matrix, with joint probability distributions given by Eqs. (6) and (7) for the quenched and annealed models, respectively. The free energy and other thermodynamic quantities, we believe, can be computed exactly (to leading orders) through the saddle point approximation. However, in the annealed model, fluctuations of the respective eigenvalues can affect the thermal average of the projected spin components, particularly for the leading components in the critical region where the denominator of Eq. (12) approaches zero. This is what we aimed to illustrate in Figs. 7 and 8, highlighting the limitations of the MLE spectrum obtained.

To further minimize misunderstanding, we propose to change the title of the paper to
“Exact finite-size scaling of the maximum likelihood spectra in the quenched and annealed Sherrington-Kirkpatrick spin glass.”

Furthermore, the following sentence is added towards the end of the abstract:

“Deviation of scaling amplitudes from their predicted values is observed in the critical region, attributable to eigenvalue fluctuations in the annealed case.”

Reviewer 3:

I believe the authors have given too few details about the numerical simulations.

For example, it is not clear whether they compute the eigenvectors of the coupling matrix J (which evolved during the numerical simulation) at each Monte Carlo step to compute the components.

Reply:

We thank the reviewer for pointing out the lack of numerical details. In the revised manuscript, the following paragraph is provided in Section 4.2:

“We performed 200 independent simulations for each system size and temperature. Data collected included spin configurations and the coupling matrix, sampled every N bond flip attempts to determine equilibrium properties.”

The eigenvectors are calculated every N attempted bond flips to determine the spin components. We have added further clarification in Section 4.2, in the paragraph that begins with “Figure 6 presents…”:

“To compute the spin components, we projected the spins onto the eigenvectors of the instantaneous coupling matrix.”

Reviewer 3:

More importantly, the authors are using a fixed ratio between the number of spin flips and bond flips: 1 bond flip for every N spin flips. How can they be sure this ratio would sample from the annealed measure?

I suggest the authors perform a more extensive set of numerical simulations where M bond flips are made for every N spin flips: if they observe the same results for any positive M value, then this a good evidence the annealed measure is sampled.
Remind that for M=0 one is sampling the quenched measured, so M=1 (the value used by the authors) can be too small and provide results from a sort of mixed ensemble.

Reply:

We thank the reviewer for noting the effects of spin-J update ratios. We have previously tested these ratios and confirmed that they do not affect the equilibrium properties of the system under the Metropolis algorithm, as long as the simulation length greatly exceeds the relaxation time. In a sequel to the current paper, we will discuss the temporal correlations of spins, as well as the eigenvalues and eigenvectors. Essentially, the memory from the slow dynamics of the J_ij’s fades when every bond has a chance to flip. The equilibration and sampling times used in our simulations are 10 and 40 times the relaxation time, respectively.

To clarify the issue raised by the reviewer from an empirical viewpoint, we have included the following sentence at the end of the third paragraph in Section. 4.2:

“Additionally, we tested varying the number of spin flips between successive bond flip attempts, from N/16 to 4N, and found consistent results across different setups.”

---

## Round 2 · Author Response

Dear Editor,

We are pleased to submit our revised manuscript for your review. Below, I've included for you detailed responses to the referees' comments, along with a summary of the changes made to the article.

Response to Reviewer 1’s Report
We thank Reviewer 1 for the positive recommendation of our manuscript. The comments on previous related work are extremely valuable. We have studied these papers and referenced them in the revised manuscript, as explained in the itemized reply below.

Reviewer 1:

Still, I believe a few references to earlier literature should be added to the text, to put it in appropriate context. In particular, I can not fully agree with the statement in the end of the paper “The discrete set of eigenvalues at the leading edge of the spectrum can then be resolved exactly to order N^(-2/3), which appears to have been an open problem before the reported work”. Not only a lot is known about the mean positions, but also fluctuations of the individual eigenvalues (in particular, the whole gap distribution between two largest/smallest eigenvalues) has been addressed for a related case of GUE matrices, see e.g.

Anthony Perret, Gregory Schehr, “Near-Extreme Eigenvalues and the First Gap of Hermitian Random Matrices,” J Stat Phys (2014) 156:843–876

and also a lot is known about extreme GOE eigenvalues.

Reply

We have revised our statement in the concluding section and added additional references as follows:

“Although the scaling of the edge spectral gaps against N, including aspects of their fluctuation behavior, has been extensively reported in the literature (see., e.g.,
[8], [25], [26]), the method we propose here uses only elementary algebra and goes beyond equilibrium solution of the usual Coulomb gas problem where the nodal points of the Airy function appear [7, 24].”

[7] L. A. Pastur and M. Shcherbina, Eigenvalue distribution of large random matrices, 171. American Mathematical Soc. (2011).
[8] S. N. Majumdar and G. Schehr, Top eigenvalue of a random matrix: large deviations and third order phase transition, Journal of Statistical Mechanics: Theory and Experiment 2014(1), P01012 (2014).
[24] Y. Chen and M. E. H. Ismail, Ladder operators and differential equations for orthogonal polynomials, Journal of Physics A: Mathematical and General 30(22), 7817 (1997).
[25] A. Perret and G. Schehr, Near-extreme eigenvalues and the first gap of hermitian random matrices, Journal of Statistical Physics 156, 843 (2014).
[26] S. N. Majumdar, A. Pal and G. Schehr, Extreme value statistics of correlated random variables: a pedagogical review, Physics Reports 840, 1 (2020).

Reviewer 1:

In particular, appearance of the Airy equation (see eq. (38) for Q(u)) in describing properties of the eigenvalues at the edge of GOE is a classical result, see e.g. section 6.3 of the monograph

L. Pastur, M. Shcherbina “Eigenvalue Distribution of Large Random Matrices” (AMS, 2011))

Reply

We have included the following sentence at the end of section 3.2.2:

“We note in passing that the close connection between the Coulomb gas problem and nodes of the Airy function has also appeared in previous work in somewhat different contexts [7, 24].”

See also our reply above.

Reviewer 1:

What concerns the positions of resolvent poles/zeros of Q(u) given in eqs. (37)-(40), it reminded me the study of positions of extreme zeroes of Hermite polynomials, described in the Theorem 5.2 in p.7826 of the paper

Yang Chen and Mourad E H Ismail
“Ladder operators and differential equations for orthogonal polynomials,” J. Phys. A: Math. Gen. 30 (1997) 7817–7829

Reply

This is an interesting article, though somewhat outside the scope of present paper. We have made reference to this work (Ref. [24]) at the end of Sec. 3.2.2 as well as in the concluding section.

Reviewer 1:

Closer to the topic of the present paper, the scaling analysis at the spectral edge, section 3.2.1, looks to me overall not dissimilar to the analysis in the section IV of the paper

Finite-size critical scaling in Ising spin glasses in the mean-field regime
T. Aspelmeier, H. G. Katzgraber, D. Larson, M. A. Moore, Matthew Wittmann, and J Yeo Phys. Rev. E 93, 032123 (2016)

which I feel also deserves mentioning in the present context.

Reply

This is indeed a very interesting paper that develops a finite-size analysis based on the spin-glass susceptibility χ_l. Their treatment of the edge spectrum captures the correct physics behind scaling properties of χ_l near the glass transition. Nevertheless, Eq. (28) in their paper is only approximate and does not follow from a systematic expansion such as the one presented in this paper. In the revised version, we added the following sentence in the third paragraph of the introductory section:
“To achieve this, it is essential to resolve the spacing between successive eigenvalues of the coupling matrix at the spectral edge, which, surprisingly, has not been adequately addressed despite many closely related work and results (see, e.g., [7–9]).”
[9] T. Aspelmeier, H. G. Katzgraber, D. Larson, M. A. Moore, M. Wittmann and J. Yeo, Finite-size critical scaling in ising spin glasses in the mean-field regime, Phys. Rev. E 93, 032123 (2016).
Furthermore, we added a sentence at the end of the first paragraph in Sec. 3.2.1:

“This problem was considered previously in Ref. [9] under an approximate scheme.”

Response to Reviewer 2’s Report:
We thank Reviewer 2 for a careful reading of our manuscript and his/her positive recommendations. Below is our reply to the reviewer’s comment:

Distinction between Ising and Spherical Models:
While the physics in the annealed case is similar the low temperature physics of the quenched problem is different. The authors should probably be careful with this distinction in their discussion?
Reply

This distinction is indeed very important and should be emphasized. In the revised manuscript, we added a footnote in the second paragraph of Sec. 2.2 after Eq. (5) with a new reference:

“Strictly speaking, this procedure applies only when there is no replica symmetry breaking. In the quenched SK model, for example, discrete Ising spins give rise to a complex energy landscape in the low temperature phase that needs to be treated differently. See, e.g., Ref. [22].”

[22] C. De Dominicis and I. Giardina, Random fields and spin glasses: a field theory approach, Cambridge University Press, ISBN 9780521847834 (2006).

Furthermore, the following sentence is added in Section 2.3, after Eq. (13):

“However, in the quenched Ising SK model, the discrete spins and frustrations prevent condensation below the critical temperature, leading the system to form a spin glass instead.”

Additionally, we note that although the low-temperature physics differ, the quenched Ising and spherical models exhibit the same finite-size scaling near the critical temperature according to our results and those in the literature, e.g., T. Aspelmeier, H. G. Katzgraber, D. Larson, M. A. Moore, M. Wittmann and J. Yeo, Finite-size critical scaling in Ising spin glasses in the mean-field regime, Phys. Rev. E 93, 032123 (2016). The paragraph below is included at the end of Sec. 5 to this effect:

“The finite-size scaling results of the edge spectrum apply equally well to the spherical model with continuous spins and a matrix of quenched coupling constants. For the SK model with Ising spins, the principal component analysis fails to capture replica symmetry breaking in the low temperature spin glass phase. Nevertheless, we expect finite-size scaling properties of the SK model at criticality to be similar to those of the spherical model, making them amenable to the analysis presented in this paper.”

We have made a few other minor rewordings throughout the paper to emphasize the distinction, and we thank the reviewer wholeheartedly for the suggestion.

Response to Reviewer 3’s Report:
We appreciate the reviewer's careful reading of our manuscript and his/her valuable feedback. The questions and comments provided are pertinent and have helped us improve the clarity of the paper. Below, we provide a detailed response to the points raised.

Reviewer 3:

…the authors claim to have an exact formula for the finite-size spectrum as the title announces. However, when compared with Monte Carlo simulations, the analytical results differ from the numerical estimates.

So, either there are problems in how the numerical simulations are carried out, or the analytical result obtained by the authors is not exact but approximate.

This needs to be clarified before the manuscript can be published.

In case the analytical formula is not exact, the manuscript needs to be revised (starting
from the title) and the hypotheses leading to an approximate result need to be stressed.

The other possibility is that Monte Carlo simulations are not measuring exactly what the authors are computing from the analytical formulas.

Reply:

We apologize for any confusion caused by the word “exact” in the title of the paper, which may seem contradictory to Figs. 7 and 8. The exact finite-size scaling refers to the maximum-likelihood estimation (MLE) of the spectrum of the coupling constant matrix, with joint probability distributions given by Eqs. (6) and (7) for the quenched and annealed models, respectively. The free energy and other thermodynamic quantities, we believe, can be computed exactly (to leading orders) through the saddle point approximation. However, in the annealed model, fluctuations of the respective eigenvalues can affect the thermal average of the projected spin components, particularly for the leading components in the critical region where the denominator of Eq. (12) approaches zero. This is what we aimed to illustrate in Figs. 7 and 8, highlighting the limitations of the MLE spectrum obtained.

To further minimize misunderstanding, we propose to change the title of the paper to
“Exact finite-size scaling of the maximum likelihood spectra in the quenched and annealed Sherrington-Kirkpatrick spin glass.”

Furthermore, the following sentence is added towards the end of the abstract:

“Deviation of scaling amplitudes from their predicted values is observed in the critical region, attributable to eigenvalue fluctuations in the annealed case.”

Reviewer 3:

I believe the authors have given too few details about the numerical simulations.

For example, it is not clear whether they compute the eigenvectors of the coupling matrix J (which evolved during the numerical simulation) at each Monte Carlo step to compute the components.

Reply:

We thank the reviewer for pointing out the lack of numerical details. In the revised manuscript, the following paragraph is provided in Section 4.2:

“We performed 200 independent simulations for each system size and temperature. Data collected included spin configurations and the coupling matrix, sampled every N bond flip attempts to determine equilibrium properties.”

The eigenvectors are calculated every N attempted bond flips to determine the spin components. We have added further clarification in Section 4.2, in the paragraph that begins with “Figure 6 presents…”:

“To compute the spin components, we projected the spins onto the eigenvectors of the instantaneous coupling matrix.”

Reviewer 3:

More importantly, the authors are using a fixed ratio between the number of spin flips and bond flips: 1 bond flip for every N spin flips. How can they be sure this ratio would sample from the annealed measure?

I suggest the authors perform a more extensive set of numerical simulations where M bond flips are made for every N spin flips: if they observe the same results for any positive M value, then this a good evidence the annealed measure is sampled.
Remind that for M=0 one is sampling the quenched measured, so M=1 (the value used by the authors) can be too small and provide results from a sort of mixed ensemble.

Reply:

We thank the reviewer for noting the effects of spin-J update ratios. We have previously tested these ratios and confirmed that they do not affect the equilibrium properties of the system under the Metropolis algorithm, as long as the simulation length greatly exceeds the relaxation time. In a sequel to the current paper, we will discuss the temporal correlations of spins, as well as the eigenvalues and eigenvectors. Essentially, the memory from the slow dynamics of the J_ij’s fades when every bond has a chance to flip. The equilibration and sampling times used in our simulations are 10 and 40 times the relaxation time, respectively.

To clarify the issue raised by the reviewer from an empirical viewpoint, we have included the following sentence at the end of the third paragraph in Section. 4.2:

“Additionally, we tested varying the number of spin flips between successive bond flip attempts, from N/16 to 4N, and found consistent results across different setups.”

---

## Round 2 · List of Changes

In addition to those listed above, we have slightly modified the abstract to reflect the changes made in the main text.

---

## Editorial Decision

published